# Crystal structure of SARS-CoV-2 Orf9b in complex with human TOM70 suggests unusual virus-host interactions

Xiaopan Gao [1,2,4], Kaixiang Zhu[1,4], Bo Qin[1,2,4], Vincent Olieric [3], Meitian Wang [3] & Sheng Cui [1,2 ✉]

Although the accessory proteins are considered non-essential for coronavirus replication, accumulating evidences demonstrate they are critical to virus-host interaction and pathogenesis. Orf9b is a unique accessory protein of SARS-CoV-2 and SARS-CoV. It is implicated in immune evasion by targeting mitochondria, where it associates with the versatile adapter TOM70. Here, we determined the crystal structure of SARS-CoV-2 orf9b in complex with the cytosolic segment of human TOM70 to 2.2 Å. A central portion of orf9b occupies the deep pocket in the TOM70 C-terminal domain (CTD) and adopts a helical conformation strikingly different from the β-sheet-rich structure of the orf9b homodimer. Interactions between orf9b and TOM70 CTD are primarily hydrophobic and distinct from the electrostatic interaction between the heat shock protein 90 (Hsp90) EEVD motif and the TOM70 N-terminal domain (NTD). Using isothermal titration calorimetry (ITC), we demonstrated that the orf9b dimer does not bind TOM70, but a synthetic peptide harboring a segment of orf9b (denoted C-peptide) binds TOM70 with nanomolar $K_D$. While the interaction between C-peptide and TOM70 CTD is an endothermic process, the interaction between Hsp90 EEVD and TOM70 NTD is exothermic, which underscores the distinct binding mechanisms at NTD and CTD pockets. Strikingly, the binding affinity of Hsp90 EEVD motif to TOM70 NTD is reduced by ~29-fold when orf9b occupies the pocket of TOM70 CTD, supporting the hypothesis that orf9b allosterically inhibits the Hsp90/TOM70 interaction. Our findings shed light on the mechanism underlying SARS-CoV-2 orf9b mediated suppression of interferon responses.

[1] NHC Key Laboratory of Systems Biology of Pathogens, Institute of Pathogen Biology, Chinese Academy of Medical Sciences and Peking Union Medical College, Beijing, China. [2] Sanming Project of Medicine in Shenzhen, National Clinical Research Center for Infectious Diseases, Shenzhen Third People's Hospital, Southern University of Science and Technology, Shenzhen, China. [3] Swiss Light Source Paul Scherrer Institut, Villigen, PSI, Switzerland. [4] These authors contributed equally: Xiaopan Gao, Kaixiang Zhu, Bo Qin. ✉email: cui.sheng@ipb.pumc.edu.cn

The ongoing Coronavirus Disease 2019 (COVID-19) pandemic has caused a Once-in-a-Century global crisis[1]. Despite scientists worldwide racing to develop antiviral drugs, curative treatments are unavailable at the time of writing. The global economy is experiencing the worst plunge in recent history amid fears of further deterioration of the COVID-19 situation.

The International Committee on Taxonomy of Viruses (ICTV) officially named the causative agent of COVID-19, severe acute respiratory syndrome coronavirus 2 (SARS-CoV-2), based on its similarity to SARS-CoV[2]. While SARS-CoV-2 and SARS-CoV share many proteins common in other CoVs, including 4 major structural proteins (S, E, M, and N proteins) and 16 nonstructural proteins (nsp1-16), they possess a unique set of proteins, namely orf3a, 3b, 6, 7a, 7b, 8a, 8b, and 9b[3,4]. Since these proteins were believed non-essential for virus replication, they have named the accessory proteins. However, this name is misleading. Many studies have demonstrated the accessory proteins are critical to the virus's survival in the host and contribute significantly to pathogenesis[4,5]. They participate in a variety of virus-host interactions ranging from cell proliferation, programmed cell death, cytokine production to antiviral immunity evasion. Disrupting virus-host interaction critical to the viral life-cycle represents a good strategy for drug design because it can avoid resistance commonly induced by direct-acting antiviral drugs[6–8]. Elucidating virus-host interaction at the molecular level is therefore fundamental to identify drug targets in the host.

The accessory protein orf9b is present in both SARS-CoV-2 and SARS-CoV. This 98-amino acid (aa) protein is encoded by an alternative open reading frame (ORF) within the N gene and is translated via a leaky scanning mechanism during translation[9]. Crystal structures of orf9b alone revealed a homodimeric β-strand-rich structure (PDB id: 2CME, 6Z4U) with a hydrophobic central tunnel for lipid binding, consistent with the role of orf9b in the mature virion assembly[10]. In hosts, SARS-CoV orf9b targets the MAVS/TRAF3/TRAF6 signalosome on mitochondria during infection and suppresses innate immunity[11]. A recent comparative viral-host protein-protein interaction analysis revealed that SARS-CoV-2 orf9b interacts with the mitochondrial outer membrane protein TOM70[12], a 70-kDa membrane-anchored adapter implicated in preprotein import into mitochondria, endoplasmic reticulum (ER)-mitochondria contacts, and the activation of antiviral signaling cascade[13]. The binding of SARS-CoV-2 orf9b to human TOM70 can lead to the suppression of interferon responses[14].

TOM70 is a multifunctional protein anchored to the mitochondrial outer membrane. It is a surface receptor of the translocase of the outer membrane (TOM) complex, the gateway of protein import in mitochondria[15]. TOM70 recognizes a group of preproteins with an internal targeting sequence and cooperates with the molecular chaperone heat shock protein (Hsp) 90 to transfer preproteins to mitochondria[16,17]. Intriguingly, TOM70 also plays a crucial role in the activation of antiviral immune responses in hosts. It is a key adapter that relays antiviral signaling from the mitochondrial antiviral signaling protein (MAVS) to TANK-binding kinase 1 (TBK1)/interferon regulatory factor 3 (IRF3)[13]. Specifically, virus infection triggers the interaction between TOM70 and MAVS. The N-terminal clamp domain of TOM70 (also referred to as the N-terminal tetratricopeptide repeats, TPRs) binds the C-terminal EEVD motif of Hsp90, resulting in the recruitment of the Hsp90/TBK1/IRF3 complex to mitochondria. Ultimately, IRF3 is phosphorylated and translocated to the nucleus activating the antiviral gene transcription.

Gordon and colleagues provided convincing evidence that human TOM70 binds both SARS-CoV and SARS-CoV-2 orf9b[12].

They showed that orf9b colocalized with mitochondria and TOM70 was identified as a high-confidence interactor of orf9b during the mapping of virus-host protein interactions. Furthermore, the authors conducted co-immunoprecipitation experiments to demonstrate that endogenous TOM70 was precipitated in the presence of orf9b. Finally, they co-purified both proteins following overexpression in *Escherichia coli* and obtained a stable TOM70-orf9b complex, which provided the foundation for structural characterization.

The cryo-electron microscopy (EM) structure of human TOM70 in complex with SARS-CoV-2 orf9b was recently determined to ~3.1 Å resolution[12]. The structure revealed that orf9b occupies the hydrophobic pocket on the C-terminal TPRs of TOM70. Given that this pocket is responsible for recognizing the internal mitochondrial targeting signal (MTS) of preproteins, occupying this site may undermine the function of TOM70. Notably, when bound to TOM70, orf9b exhibited a helical conformation in stark contrast to the β-strand-rich structure of the orf9b homodimer in the absence of protein binding partner[10]. This unusual helical conformation of orf9b is reminiscent of a study demonstrating that most mitochondrial targeting sequence tends to form amphiphilic helices[18].

TOM70 harbors two distinct sites for protein-protein interaction; while the N-terminal TPRs of TOM70 associate with the heat shock protein family molecular chaperones, the C-terminal TPRs bind mitochondrial preproteins for import. Whether substrate-binding at one site affects the binding at another site remains to be fully elucidated. A fluorescence anisotropy experiment showed that yeast TOM70 could bind the peptide corresponding to the extreme C-terminal segment of Hsp70 (last eight residues) and the peptide derived from a precursor protein (or preprotein) MTS sequence (181–193aa of the yeast mitochondrial phosphate carrier protein)[19]. Intriguingly, the binding affinity of yeast TOM70 to the precursor protein-peptide was unchanged in the presence of the Hsp70 peptide, suggesting that binding at the N-terminal TPRs has little effect on binding at the C-terminal TPRs. Li and colleagues determined crystal structures of TOM71 (a TOM70 homolog that also mediates preprotein transfer) complexed with the EEVD motif of the molecular chaperones Hsp70/Hsp90. Structural characterization suggested that upon binding to the EEVD motif, TOM71 was fixed in an "open state" in which the pocket receiving preprotein adopted a favorable conformation for the loading of preproteins[20].

Although a cryo-EM structure of the human TOM70/SARS-CoV-2 orf9b complex has been determined to a reasonable resolution, it is challenging to further improve the resolution using this approach. By contrast, crystallographic methods can often achieve higher resolution structures and provide further details. In addition, robust crystallization of human TOM70 is important to structure-based drug design targeting this host protein. For example, fragment-based screening (FBS) has been widely applied for identifying novel lead compounds against SARS-CoV-2 since the outbreak of Covid-19[21,22]. High-quality crystals of drug target are usually prerequisite for the FBS method.

In this study, we determined the crystal structure of the human TOM70/SARS-CoV-2 orf9b complex to 2.2 Å resolution. Orf9b occupies the hydrophobic pocket on the TOM70 C-terminal domain (CTD). Owing to the high resolution of this structure, we identified 12 hydrogen bonds and 7 salt bridges between orf9b and TOM70, as well as 23 non-polar residues of TOM70 that recognize orf9b. We assessed the interaction between TOM70 and orf9b using isothermal titration calorimetry (ITC) and found that while the orf9b homodimer did not bind TOM70, a synthetic peptide corresponding to a central segment of orf9b (denoted the C-peptide) bound TOM70 with nanomolar $K_D$. Furthermore, we

demonstrate that the binding affinity between the Hsp90 C-terminal EEVD motif and TOM70 was greatly reduced when TOM70 was associated with orf9b. Similarly, the C-peptide also blocked the binding of the EEVD motif to TOM70, although to a lesser extent. Conversely, pre-incubating TOM70 with the EEVD motif has little effect on the binding affinity of the C-peptide to TOM70. In summary, our findings support the hypothesis that orf9b allosterically inhibits binding of the molecular chaperon Hsp90 to TOM70.

## Results

**Assembly of the human TOM70/SARS-CoV-2 orf9b complex.** To gain structural insight into the interaction between TOM70 and orf9b, we sought to determine the crystal structure of the cytoplasmic segment of human TOM70 (hTOM70) in complex with SARS-CoV-2 orf9b (Fig. 1a, b). Surprisingly, by mixing separately purified hTOM70 and SARS-CoV-2 orf9b did not yield a complex. Given that orf9b self-dimerizes, the orf9b dimer might prevent the interaction with hTOM70. We, therefore, employed a co-expression strategy: genes of a selection of hTOM70 variants and SARS-CoV-2 orf9b were inserted into multiple cloning sites (MCS)1 and 2 of the pETDuet-1 vector, respectively. While no affinity tag was appended to hTOM70, we fused an N-terminal His-tag to orf9b. We found that three hTOM70 variants with progressively shortened N-terminal portions, hTOM70 ΔN1 (60–608), hTOM70 ΔN2 (106–608), and hTOM70 ΔN3 (235–608), co-eluted with SARS-CoV-2 orf9b (Fig. 1b), indicating the successful complex formation. We performed crystallization screening for all complexes, and only hTOM70 ΔN2/SARS-CoV-2 orf9b yielded measurable crystals. We then accessed the stoichiometry of this complex using size-exclusion chromatography (SEC) and analytical ultracentrifugation (AUC; Fig. 1c, d). The calculated molecular mass of SARS-CoV-2 orf9b alone was ~21 kDa in either SEC or AUC, almost twice the theoretical molecular mass of a monomer (10.8 kDa). The calculated molecular mass of hTOM70 ΔN2 was ~55 kDa, matching the size of the theoretical monomer (56.9 kDa). The molecular mass of the hTOM70 ΔN2/SARS-CoV-2 orf9b complex was calculated ~68 kDa, matching the sum of 1 hTOM70 ΔN2 monomer plus 1 orf9b monomer (67.9 kDa), indicating the stoichiometry of the complex is 1:1.

**Crystal structure of the human TOM70-Orf9b complex.** We determined the crystal structure of the hTOM70 ΔN2/SARS-CoV-2 orf9b complex to 2.2 Å. One copy of TOM70 and one copy of orf9b are present in the asymmetric unit, consistent with our stoichiometry assessment. While the majority of hTOM70 residues were visible in our final structure, we found a central portion of SARS-CoV-2 orf9b (residues 43-78) was bound to hTOM70, but the remnant of orf9b was disordered. Similar to yeast TOM70/TOM71 (PDB id: 2GW1 and 3FP3), the hTOM70 cytosolic domain is comprised of 25 tightly packed helices (α1-25), of which 22 helices form 11 tetratricopeptide repeats (TRP 1-11) (Fig. 2a left). The structure can be divided into the N-terminal domain (NTD, α1-7) harboring the pocket recognizing the "EEVD" motif of the molecular chaperones Hsp70/90, denoted the NTD-pocket and the C-terminal domain (CTD, α8-25) harboring the pocket recognizing the internal mitochondrial targeting signals (MTS) of preprotein, denoted the CTD-pocket. Three special helices α7, α8, and α25 do not form TPR; instead, α7 bridges the NTD-CTD, α8 constitutes the bottom of the CTD-pocket and α25 forms a tall wall of the CTD-pocket.

We calculated the composite omit map of this structure which unambiguously showed that a 35 aa fragment of SARS-CoV-2 orf9b accommodates the CTD-pocket of hTOM70 (Fig. 2a right).

This is consistent with the function of TOM70 in recognizing an internal fragment of preproteins. The visible portion of SARS-CoV-2 orf9b adopts a helical conformation: The N-terminal region of the fragment folds into a "U-turn" shaped loop that is followed by a long, slightly bent helix and a short helix; the C-terminal region of the fragment extends towards the proximity of hTOM70 NTD. This is in stark contrast to the β-sheet-rich conformation adopted by the same fragment in the orf9b homodimer (Fig. 2b and Supplementary Fig. 1). Using PISA software, we calculated the buried interface area between hTOM70 and SARS-CoV-2 orf9b of ~2000 Å$^2$ and the solvation free energy gain (Δ$^i$G) of −33.8 kcal/M. Owing to the high resolution of this structure, we were able to identify extensive interactions between the two proteins: (1) Polar contacts include 12 potential hydrogen bonds and 7 potential salt bridges, involving 10 residues of hTOM70 (from α13, α19, α21, α23, and α25 helices) and 11 residues of SARS-CoV-2 orf9b (Fig. 2c–e). Of note, multiple salt bridges (distance 2.7–3.3 Å) were identified between E549, H583 of hTOM70, and R58, E65 of orf9b, suggesting their importance in the interaction. While the majority of those polar residues on the surface of the CTD pocket are conserved, the counterpart of E549 in yeast TOM70/TOM71 is isoleucine. (2) We identified 23 non-polar residues on the surface of the CTD pocket interacting with SARS-CoV-2 orf9b. While most are conserved, residues F256, F546, and T553 of hTOM70 are replaced by hydrophilic residues S268, Q565, and Q572 in yeast TOM70/TOM71(Supplementary Fig. 2). The presence of non-conserved residues inside the CTD pocket implies differences in MTS substrate specificity in higher organisms.

**Orf9b binding stabilizes the conformation of TOM70.** Because the unliganded hTOM70 structure remains unavailable, we superimposed our structure to yeast TOM70/TOM71 structures to deduce conformational changes associate with orf9b binding, assuming the structure of hTOM70 resembles the yeast homologs. The structure of hTOM70 in the presence of orf9b is similar to yeast TOM71 structures, all of which adopt an "opened" NTD. However, the NTD in the "closed" conformation observed in the unliganded yeast TOM70 would clash with orf9b (Fig. 3a, b), suggesting that orf9b binding favors the open conformation. One prominent structural rearrangement takes place in a region starting from helix α7 bridging NTD-CTD to helix α8 located at the bottom of the CTD pocket. We, therefore, denoted helices α7 and α8 (as well as their counterparts in yeast TOM70/TOM71) as the "bridging helix" and the "bottom helix", respectively. A long loop (~35 aa) connecting the bridging helix and the bottom helix is disordered in yeast TOM70/TOM71 structures, suggesting intrinsic flexibility of this region. Comparing the structure of hTOM70/orf9b complex and the unliganded structures indicates that the binding of orf9b induces evident conformational changes, which includes movement of the bridging helix outward or downward compared with the corresponding helix in yeast TOM70 or TOM71 structure, and the bottom helix is pressed down by the N-terminal portion of the orf9b (Fig. 3c). It is intriguing that in the presence of orf9b, the loop between the bridging helix and the bottom helix becomes visible as if the α7-to-α8 loop has been pulled straight and stabilized. Collectively, we identified a highly flexible region of the CTD-pocket of hTOM70 associated with orf9b binding. By occupying the CTD pocket, the orf9b may lock the conformation of hTOM70 into a rigid state.

**Cooperative binding at the NTD- and CTD-pockets.** It has been shown that binding of the EEVD motif at the NTD-pocket of TOM70 allosterically promotes substrate binding at the CTD-

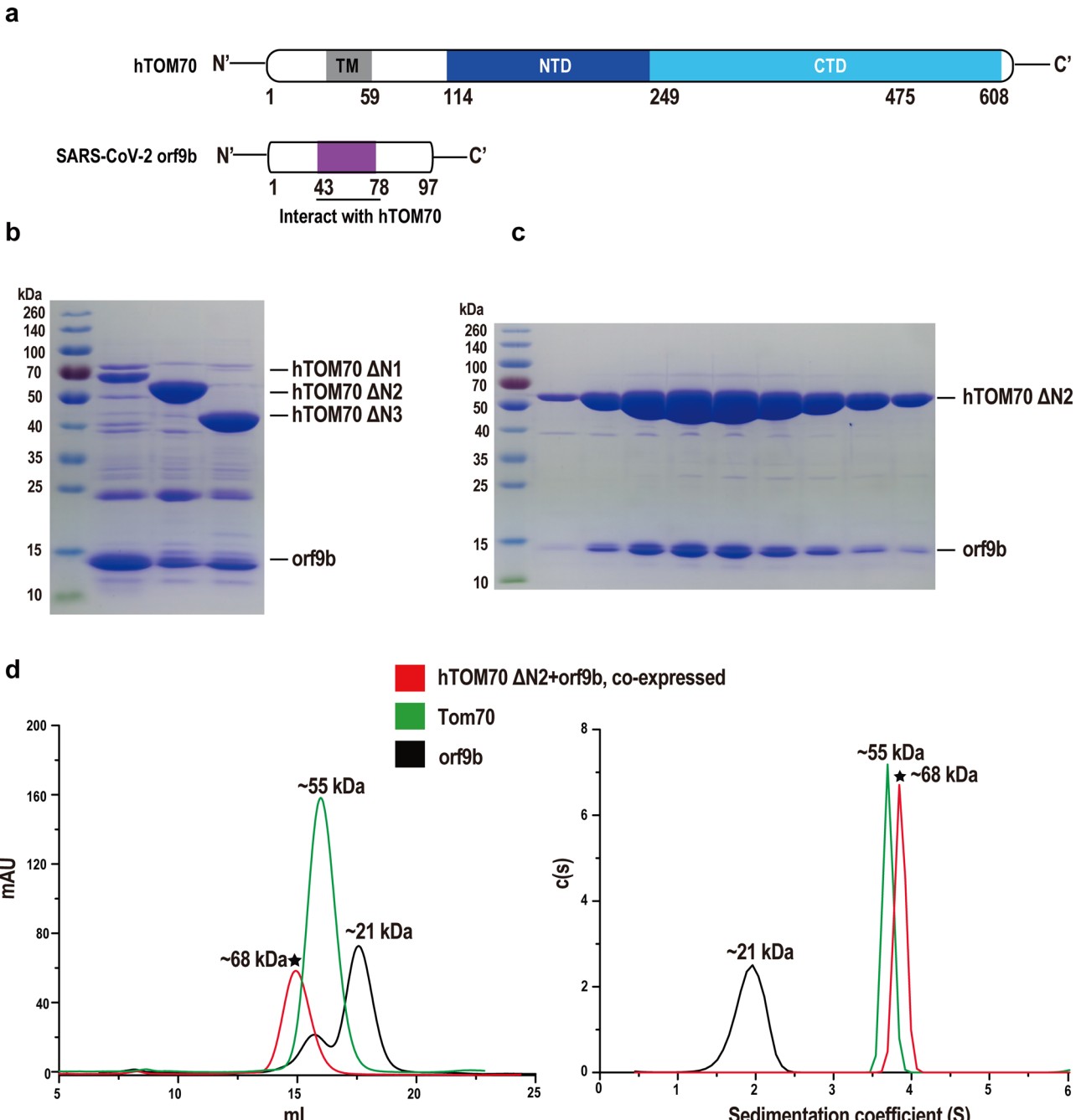

**Fig. 1 Expression and assembly of the hTOM70/SARS-CoV-2 orf9b complex. a** Diagrams of SARS-CoV-2 orf9b and hTOM70 proteins. TM transmembrane domain, NTD N-terminal domain, CTD C-terminal domain. **b** When co-expressed in *E coli*, three of hTOM70 truncations, hTOM70 ΔN1 (60-608), hTOM70 ΔN2 (106-608), and hTOM70 ΔN3 (235-608) without fusion tag co-elute with SARS-CoV-2 orf9b with an N-terminal His-tag. **c** SDS-PAGE analysis of fractions eluted from gel filtration column, demonstrating hTOM70 ΔN2 and SARS-CoV-2 orf9b formed stable complex. **d** Left, size-exclusion chromatography and right, analytical ultracentrifugation analyses three samples: co-expressed SARS-CoV-2 orf9b/hTOM70 ΔN2 complex (red), alone expressed hTOM70 (green) and SARS-CoV-2 orf9b (homodimeric, black). The calculated molecular weight is indicated on top of the peaks. Data for SARS-CoV-2 orf9b/hTOM70 ΔN2 complex are indicated with stars.

pocket[20]. Conversely, the impact of orf9b binding at the CTD-pocket on the NTD-pocket remained to be understood. We first analyzed the surface electrostatic potential of hTOM70 (Fig. 4a). The NTD-pocket is a solvent assessable and positively charged clamp, whereas the CTD pocket is a hydrophobic and negatively charged deep cavity. This suggests that the mechanisms underlying substrate recognition at NTD and CTD must be different. We modeled the binding of the EEVD motif to hTOM70 by superimposing our structure onto the structure of the yeast TOM71/EEVD motif complex (PDB id: 3FP2). The superimposition aligned 98 Cα atoms (residue 110–208 of hTOM70) and gave a root mean square deviation (rmsd) value of 0.72 Å. It appears that the C-terminal portion of orf9b, although invisible in the crystal structure, reaches into the proximity of the NTD-pocket (Fig. 4b, c).

Using ITC, we investigated the interaction between hTOM70 and homodimeric SARS-CoV-2 orf9b, but the binding was not detected (Supplementary Fig. 3a). This is consistent with the

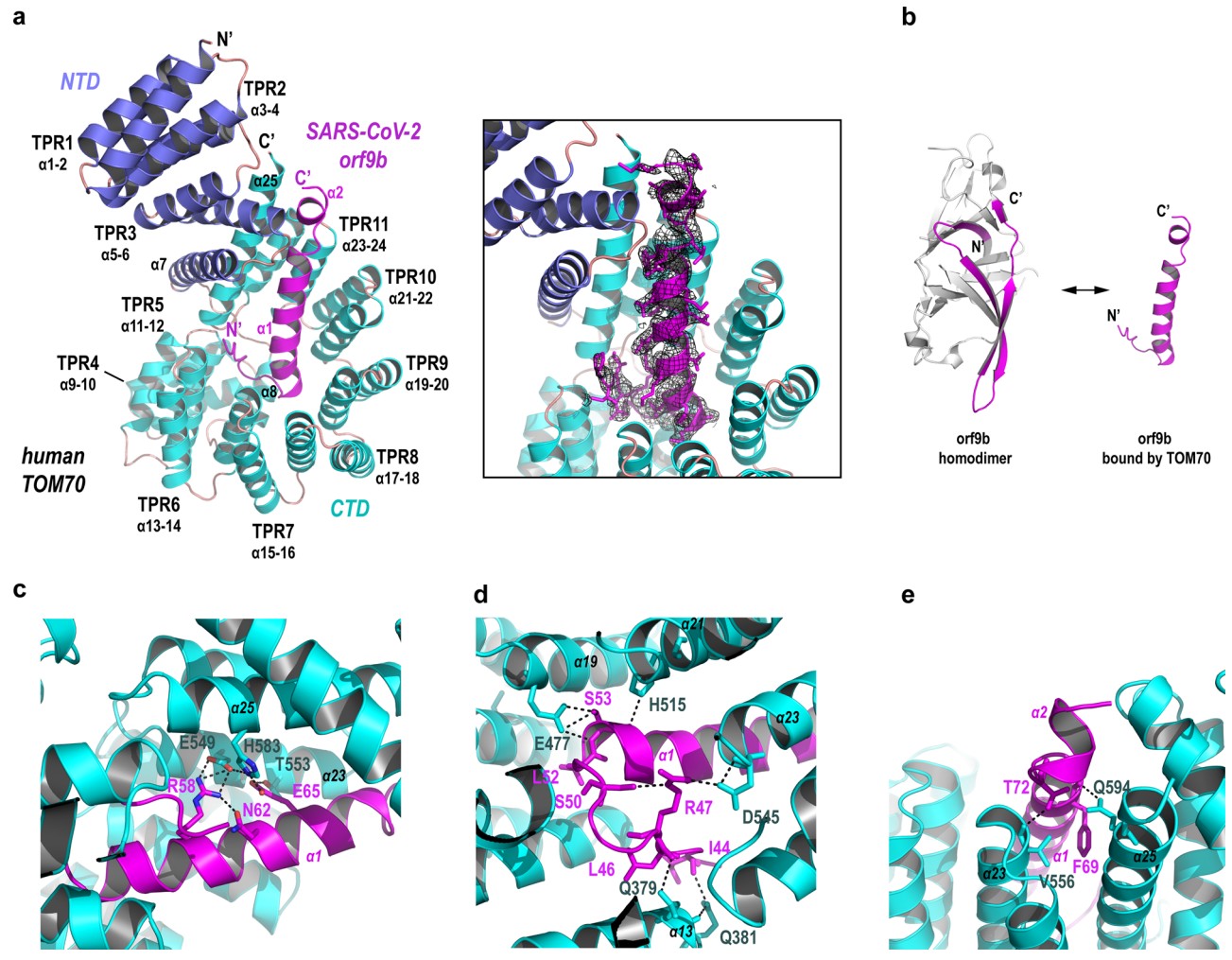

**Fig. 2 Crystal structure of the hTOM70/SARS-CoV-2 orf9b complex. a** Left, ribbon model of the hTOM70 cytosolic domain with SARS-CoV-2 orf9b occupying the CTD pocket. The NTD of hTOM70 is colored blue, the CTD is cyan and orf9b is magenta. Tetratricopeptide repeat (TPR) motifs and secondary structure elements are indicated. Right, magnified view of SARS-CoV-2 orf9b inside the CTD pocket. The omit map (black mesh, calculated by program phenix.composite_omit_map) for orf9b is superimposed on the model. **b** Ribbon models of dimeric SARS-CoV-2 orf9b and the central portion of the same protein when bound to hTOM70. The segment of SARS-CoV-2 orf9b 43–78 adopts a β-strand rich structure in its dimeric form (right) and an α-helix rich structure in its bound form (left). **c–e** Details of polar interactions between SARS-CoV-2 orf9b and hTOM70.

observation that hTOM70 and the SARS-CoV-2 orf9b dimer cannot assemble into a stable complex in vitro, which indicates that the SARS-CoV-2 orf9b dimer alone does not have the structural plasticity to transform into the helical conformation required for binding with hTOM70. Understanding factors driving the structural transformation of orf9b is therefore crucial for deciphering its function during viral infection, and this warrants future investigation.

To overcome the difficulty in investigating the interplay between orf9b and TOM70, we designed two peptides: One contains the EEVD motif of the human molecular chaperon Hsp90, denoted the N-peptide, another harbors a segment of SARS-CoV-2 orf9b (44–70) that was found inside of the CTD-pocket of hTOM70, denoted the C-peptide. Under the same conditions, we found that interaction between the unliganded hTOM70 and the N-peptide was an exothermic and enthalpically-driven process, with $\Delta H =$ −4150 cal/mole, $\Delta S = 11.7$ cal/mole/deg and $K_D = 2.56$ μM; by contrast, the interaction between the unliganded hTOM70 and the C-peptide was an endothermic and entropically-driven process, with $\Delta H = 2651$ cal/mole, $\Delta S = 36.4$ cal/mole/deg and $K_D = 0.96$ μM (Fig. 4d, e, Supplementary Table 2). The distinct thermodynamic features of binding at the two pockets of TOM70

are in line with our structural analyses: the NTD-pocket/EEVD interaction is mainly through electrostatic attraction, whereas the CTD-pocket/orf9b interaction involves more hydrophobic interactions and hydrogen bonds than salt-bridges.

Next, we titrated the synthetic peptides against the hTOM70/SARS-CoV-2 orf9b complex prepared via co-expression. We observed negligible enthalpy changes when the C-peptide was titrated into the hTOM70/SARS-CoV-2 orf9b complex, indicating little or no binding (Fig. 4d). Possibly, this short fragment was unable to compete with the full-length protein that had already occupied the CTD pocket. Strikingly, the binding affinity of the N-peptide to the hTOM70/SARS-CoV-2 orf9b complex reduced by ~29 folds compared with the binding affinity of the N-peptide to the unliganded hTOM70 (Fig. 4e). To investigate whether the C-peptide had a similar effect in blocking N-peptide binding, we titrated the N-peptide into the hTOM70/C-peptide complex. The C-peptide was pre-incubated with the unliganded hTOM70 (molar ratio 3:1) to allow complex formation, and the N-peptide was then added to the complex. We observed ~13 folds reduction in binding affinity of the N-peptide to the hTOM70/C-peptide complex (Supplementary Fig. 3b) compared with that between the N-peptide and the unliganded hTOM70. The C-peptide was

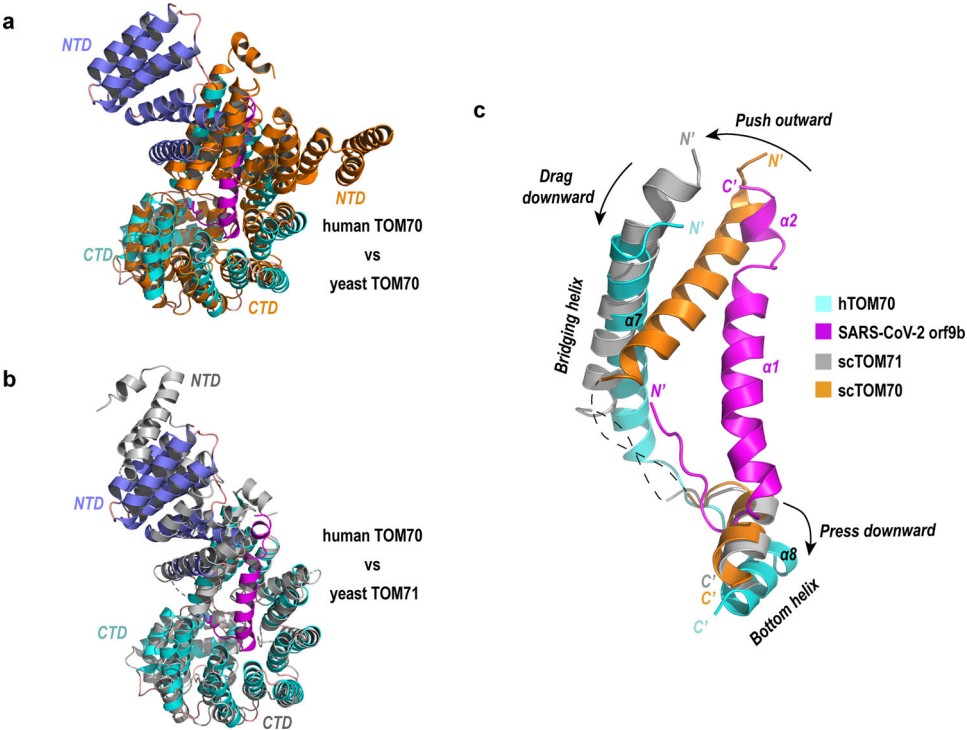

**Fig. 3 Orf9b binding induced conformational rearrangement. a** Superimposition of the hTOM70/SARS-CoV-2 orf9b complex with yeast TOM70 (PDB id: 2GW1). The NTD of hTOM70 is colored blue, the CTD is colored cyan. SARS-CoV-2 orf9b is colored magenta. **b** Superimposition of the hTOM70/SARS-CoV-2 orf9b complex with yeast TOM71 (PDB id: 3FP3). hTOM70 and SARS-CoV-2 orf9b are colored as in panel **a**. **c** Superimposition of the hTOM70/SARS-CoV-2 orf9b complex with yeast TOM70 and TOM71. Only the bridging helices and the bottom helices are shown to illustrate conformational rearrangement induced by SARS-CoV-2 orf9b binding.

clearly able to block N-peptide binding similar to the intact orf9b, albeit to a lesser extent. The difference is likely attributed to the significantly lower molecular mass of the C-peptide. Conversely, we also titrated the C-peptide to the hTOM70/N-peptide complex. The N-peptide was pre-incubated with unliganded hTOM70 (molar ratio 3:1) to allow complex formation, and the C-peptide was then added. The binding of the C-peptide to the hTOM70/N-peptide complex was also an endothermic process with $K_D = 0.42\,\mu M$ (Supplementary Fig. 3c). This result indicates that the N-peptide has little effect on the binding affinity of the C-peptide to hTOM70. Collectively, our findings suggest that binding of orf9b at the CTD-pocket negatively affects the binding of the EEVD motif at the NTD-pocket. In other words, orf9b allosterically inhibits substrate binding at the NTD-pocket.

Finally, we investigated the binding of the C-peptide to a selection of hTOM70 mutants (Supplementary Fig. 3d–h). We disrupted specific polar contacts between SARS-CoV-2 orf9b and hTOM70 (Fig. 2C–E) by introducing mutation E477A, E549I, E549A, D545A, and H583Q to hTOM70, respectively. Most mutants exhibited reduced binding affinity to the C-peptide at various levels. Of note, all mutants bound the C-peptide in an endothermic manner, similar to the wild-type interaction. Mutant E477A showed ~267 folds decrease in affinity compared with the wild-type protein, and mutant D545A showed ~52 folds decrease in affinity, suggesting critical roles of these residues in substrate recognition. It is not surprising that both residues are highly conserved in human and yeast TOM70/TOM71 homologs (Supplementary Fig. 2). By contrast, although E549 forms salt-bridges with R58 of orf9b, mutations E549I and E549A only had a moderate impact on binding. H583 forms a salt bridge with E65 of orf9b, but mutant H583Q exhibited nearly the same binding affinity ($K_D = 0.99\,\mu M$) as wild-type TOM70. Notably, the counterparts of E549 and H583 in yeast TOM70/TOM71 are

isoleucine and glutamine. Thus, non-conserved residues appear to play less important roles in substrate recognition than conserved residues.

### Comparison of cryo-EM and crystal structures of the TOM70/orf9b complex.
Integrating complementary structural methods is ideal for obtaining unbiased and comprehensive structural insights into the interplay between hTOM70 and SARS-CoV-2 orf9b. While Gordon and colleagues determined the cryo-EM structure of the 68 kDa hTOM70/SARS-CoV-2 orf9b complex to an impressive 3.1 Å resolution[12], we solved the crystal structure of the complex to a higher resolution (2.2 Å), and our structure provides further details. Superimposition of the two structures gave an overall rmsd value of 1.3 Å with >90% Cα atoms aligned (Supplementary Fig. 4). The consistency of structures determined from dispersed (single particles) and packed (crystal lattice) molecules reflects the unusual rigidity of TOM70 in the presence of orf9b. By contrast, structural characterizations reveal intrinsic flexibility of yeast TOM70/71 when the C-terminal pocket is unoccupied; the protein may adopt close or open conformations (Fig. 3a, b) to block or accept preproteins[20]. Biophysical studies also indicate that TOM70 has an unusual folding pathway and exhibits low structural stability in solution[23]. Thus, the rigidification of TOM70 is a key consequence of orf9b binding which undermines the role of TOM70 in protein-protein interactions. When probing the mechanism of orf9b binding, we found that E549 of hTOM70 forms a salt bridge with R58 of SARS-CoV-2 orf9b in our crystal structure, whereas in the EM structure, E580 rather than E549 of hTOM70 salt bridges with R58, which reflects a dynamic in electrostatic interactions between the two proteins. The N-terminal and C-terminal portions of SARS-CoV-2 orf9b outside the CTD pocket of hTOM70 are not visible in either the crystal or EM structure. It is possible that these

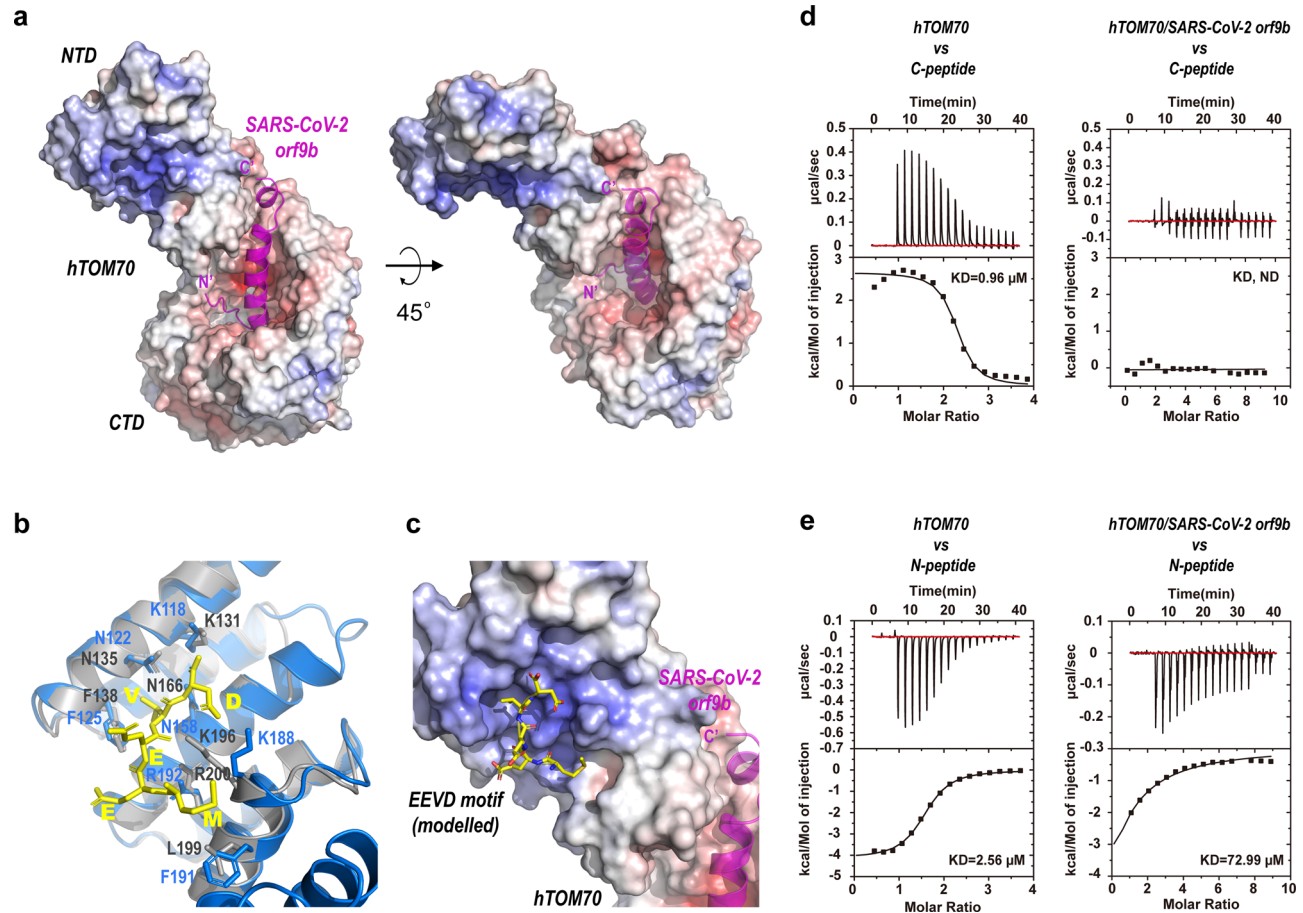

**Fig. 4 Cooperative binding at the NTD- and CTD-pockets. a** Surface electrostatic potential plot of hTOM70 with SARS-CoV-2 orf9b accommodating the CTD-pocket shown in magenta cartoon representation. **b** Superimposition of hTOM70 (blue) with the yeast TOM71-EEVD complex (PDB id: 3FP2). Yeast TOM71 is colored gray and the EEVD motif is colored yellow. **c** The EEVD motif (yellow) is modeled in the NTD-pocket of hTOM70. **d** Left, binding isotherms for the interaction of hTOM70 and the C-peptide (derived from the visible portion of SARS-CoV-2 orf9b in the crystal structure), which indicates an endothermic process; right, binding isotherms for pre-formed hTOM70/SARS-CoV-2 orf9b complex and the C-peptide, which shows little nor no binding. **e** Left, binding isotherms for the interaction of hTOM70 and the N-peptide (harboring the EEVD motif at the C-terminal region of Hsp90), which indicates an exothermic process; right, binding isotherms for pre-formed hTOM70/SARS-CoV-2 orf9b complex and the N-peptide; the binding affinity is reduced significantly.

regions may influence TOM70 functions through interactions with other proteins. Identifying the binding partners of the TOM70/orf9b core is important to fully understand the role of orf9b in infection.

## Discussion

Both SARS-CoV and SARS-CoV-2 exploit the accessory protein orf9b to target mitochondria and suppress antiviral innate immunity[11,14,24]. This conserved immune evasion pathway presents a potential antiviral target against SARS-like pathogens and possibly new variants of viruses in future outbreaks. Recent comparative virus-host protein-protein interaction screenings revealed that TOM70, a membrane-anchored adapter crucial to mitochondria functions including antiviral immune signaling, is the molecular target of orf9b[12,25].

The consequence of orf9b's association to TOM70 is the key to deciphering the immune evasion strategy adopted by this protein. Using ITC, we established a binding assay that could distinguish the binding of the EEVD motif to the NTD-pocket (also known as the N-terminal clamp-type TPR domain) from the binding of orf9b to the CTD-pocket (the C-terminal TPR domain). While the former is an exothermic process and the latter is endothermic,

and these observations are supported by our structural analyses (Fig. 4). Our mutagenesis studies also showed that although mutations at the CTD-pocket of hTOM70 weakened binding to orf9b, the endothermic signatures of those mutants in binding remained similar to those of the wild-type protein (Supplementary Fig. 3). Based on these findings, we demonstrated that occupation of SARS-CoV-2 orf9b at the CTD-pocket of hTOM70 severely disrupts binding of the EEVD motif at the NTD-pocket. We postulated a possible mechanism: (1) As implied by structural characterizations, the accommodation of orf9b at the CTD-pocket may rigidify the overall conformation of TOM70, which probably fixes the NTD-pocket in an unfavorable conformation for binding to Hsp90. (2) Although the C-terminal portion of SARS-CoV-2 orf9b is not visible in both crystal and EM structures, it is in the vicinity of the NTD-pocket where it might interfere with the binding of the EEVD motif.

In summary, we have provided experimental evidence supporting the postulation that SARS-CoV-2 orf9b allosterically inhibits the binding between hTOM70 and Hsp90, which may consequently undermine the recruitment of Hsp90 to TBK1/IRF3, and thus ultimately weakens the signaling cascade for interferon activation[13,26].

## Methods

**Reagents**. All chemicals and reagents used in this study were purchased from Sigma-Aldrich unless specified.

**Protein expression and purification**. The genes of SARS-CoV-2 orf9b and hTOM70 were synthesized and codon-optimized for expression in *E. coli* (Supplementary Table 3). Two genes were amplified by PCR and inserted into the same vector for co-expression. While the gene of SARS-CoV-2 orf9b was inserted to the ORF1 (between BamH I and Hind III restriction sites) of pETDuet-1 vector, the gene of hTOM70 truncations was inserted to the ORF2 (between Nde I and Xhol I restriction sites) (Supplementary Table 4). To prepare the unliganded hTOM70 variants, the gene was inserted into the pET28a vector (between NcoI and XhoI restriction sites), encoding the C-terminal His-tagged protein (Supplementary Table 4). The sequence of all plasmids was verified by DNA sequencing.

For protein expression. the plasmid was transformed to BL21 (DE3) competent cells. The expression was induced when the culture grew to $OD_{600}$ ~0.8. Isopropyl β-D-1-thiogalactopyranoside (IPTG, 0.5 mM final concentration) was added to the pre-cooled bacteria culture for induction. The culturing continued at 18 °C for 18 h.

To purified recombinant proteins, bacteria were harvested by centrifugation (3470×$g$, 10 min) and suspended in lysis buffer (50 mM Tris-HCl, pH = 8.0, 150 mM NaCl, 10 mM imidazole, 10 mM β-mercaptoethanol, and 1 mM PMSF). The cells were disrupted by ultrasonication. The resulting mixture was clarified by centrifugation (47,850×$g$, 50 min), and the supernatant was passed through the 0.45 μm syringe filter before loading to Ni-NTA resin pre-equilibrated with the lysis buffer. After the washing of unbound protein with 10× lysis buffer, the His-tagged proteins were eluted by the elution buffer containing 300 mM imidazole. Next, the eluate was loaded to a HiTrap Q HP column (GE Healthcare) equilibrated with the buffer containing 20 mM Tris-HCl, pH = 8.0, 75 mM NaCl, and eluted with a 10-1000 mM NaCl gradient. The fraction containing the target protein was finally purified by size-exclusion chromatography using the Superdex 200 10/300 column (GE Healthcare) pre-equilibrated with a storage buffer containing 20 mM Tris pH = 8.0, 100 mM NaCl.

**Crystallization and structure determination**. SARS-CoV-2 orf9b/TOM70 (106-608) complex was concentrated to 10 mg/ml before crystallization trials. The complex was crystallized by mixing 1 μl protein and 1 μl reservoir buffer containing 0.1 M Bis-Tris pH = 6.5, 19–25% PEG3350 in a hanging drop vapor diffusion system at 18 °C. The crystals appeared in 15–20 days and grew to maximum size in about 30 days. To freeze crystals, the crystals were transferred to the reservoir buffer supplemented with 20% ethylene glycol and soaked for 30–60 s before flash-frozen in liquid nitrogen. Complete X-ray diffraction data were collected at 100 K at the Shanghai synchrotron radiation facility (SSRF) beamline BL17U, Shanghai China. The data was processed using the XDS package[27]. The structure was solved by molecular replacement using the software Phaser MR (PDB 7KDT was used as the searching model) to yield an interpretable initial electron density map[28]. Manual model building was conducted using the software Coot and the structure was finally refined using the software PHENIX[29]. The statistics of data collection and structure refinement are summarized in Supplementary Table 1. Structural presentations were prepared using the software Pymol.

**Analytical ultracentrifugation**. Analytical ultracentrifugation was carried out using Beckman Optima XL-I equipped with an AN-50 Ti rotor with two-channel charcoal-filled centerpieces. The protein samples were freshly prepared in a buffer containing 20 mM Tris pH = 8.0, 100 mM NaCl, and concentrated to absorbance 280 nm = 0.8 before loading. The differential sedimentation coefficients, c(s), frictional coefficients, and molecular weight were calculated by the XL-I data analysis software.

**Size-exclusion chromatography**. The purified protein samples were loaded to Superdex 200 10/300 GL column (GE healthcare) pre-calibrated using molecular weight standards: γ-globulin (158 kDa), ovalbumin (45 kDa), myoglobin (17 kDa), and vitamin B12 (1.35 kDa) in gel filtration buffer containing 20 mM Tris pH = 8.0, 100 mM NaCl. The elution fractions were further analyzed with SDS-PAGE.

**Isothermal titration calorimetry**. Isothermal titration calorimetry (ITC) experiments were carried out using MicroCal iTC200 calorimeter (MicroCal, USA) at 25 °C as previously described[30,31]. The N-peptide (PLEGDDDTSRMEEVDQ, derived from human Hsp90 C-terminal portion; the EEVD motif is underlined) and the C-peptide (containing SARS-CoV-2 orf9b core sequence 44-70 aa; the sequence of this peptide and derivatives, and their applications are in patent pending) were synthesized by Scilight Biotechnology, LLC. Both the protein and peptides were dissolved in the buffer containing 20 mM Tris-HCl, pH = 8.0, and 100 mM NaCl. The concentrations of peptides in the syringe were 1 mM and the concentration of proteins in the sample cell was from 0.016-0.02 mM. 18 consecutive 2 μl injections of peptides were titrated into 350 μl sample cell with a 120 s interval between injections using a stir rate of 400 rpm. A single-site binding model was used to was used for nonlinear curve fitting using Microcal Origin software

provided by the manufacturer. ITC titration was repeated at least twice for each experiment.

**Reporting summary**. Further information on research design is available in the Nature Research Reporting Summary linked to this article.

## Data availability

The atomic coordinates and structure factors have been deposited in the Protein Data Bank under the accession codes:7DHG. Other data are available from the corresponding author upon reasonable request. Source data are provided with this paper.

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

## Acknowledgements

We thank the staff of BL17B/BL18U1/BL19U1/ BL19U2/ BL01B beamlines at National Center for Protein Science Shanghai and Shanghai Synchrotron Radiation Facility, Shanghai China, for help with data collection. This work was supported by the National Key Research and Development Program of China (2016YFD0500300 and 2019YFC0840602), National Natural Science Foundation of China (81971985, 82072291, 81772207, 11775308), Chinese Academy of Medical Science (CAMS) Innovation Fund for Medical Sciences (award Nos. 2017-I2M-1-014 and 2016-I2M-1-013), and the Guangdong Foundation for Basic and Applied Basic Research (No.2019B1515120041).

## Author contributions

S.C. and X.G. designed the study. S.C. and X.G. solved the structure and wrote the paper. X.G., B.Q., K.Z., M.W., and V.O. performed experiments, analyzed the data, and revised the paper. All authors reviewed the results and approved the final version of the manuscript.

## Competing interests

The patent protecting the design and application of the C-peptide (including its derivatives) originated from SARS-CoV-2 orf9b sequence is pending by the authors of this paper and the Institute of Pathogen Biology, Chinese Academy of Medical Sciences and Peking Union Medical College, China. The authors declare no other competing interests.
