## [Peer Review File · Nature Communications]

REVIEWERS' COMMENTS

Reviewer #1 (Remarks to the Author):

The identification and characterization of interactions between proteins of the pandemic SARS-CoV-2 coronavirus and proteins of its human host constitute important groundwork for the development of potential drugs to treat COVID-19, the disease caused by SARS-CoV-2.

In the presented manuscript, Gao et al. describe the crystal structure of a binary complex formed by the cytosolic portion of the human mitochondrial translocase TOM70 and the protein encoded by open reading frame 9b of SARS-CoV-2 (ORF9b) to a resolution of 2.2 Å. They compare their complex structure to a previously solved structure of substrate-free TOM70 from yeast and find that binding of ORF9b may stabilize TOM70 in a rigid conformation. Additionally, the authors perform ITC experiments using TOM70 or TOM70/ORF9b with synthetic peptides derived from ORF9b and from the chaperone Hsp90 as binding partners. The peptides bind to spatially distinct binding sites located in the C-terminal domain (CTD) and in the N-terminal domain (NTD), respectively. They find that the binding constant of the Hsp90-peptide and the preformed TOM70/ORF9b complex is strongly reduced compared to that of Hsp90-peptide and TOM70 alone. From this, the authors conclude that binding of ORF9b to TOM70 suppresses binding of Hsp90-peptide to the NTD of TOM70 allosterically. Furthermore, the authors determine the binding constants of several point mutants of TOM70 to ORF9b and essentially confirm the interaction interface observed in their crystal structure.

The study by Gao et al. reports a highly relevant crystal structure of the complex between the accessory protein ORF9b of SARS-CoV-2 and the host protein TOM70 that may have implications for future therapeutic approaches. The technical quality of the data seems to be generally high and the presented data support the conclusions of the manuscript. However, I have serious reservations to recommend the manuscript for publication in its current state.

Major points:

1. It is not mentioned that a cryo-EM structure of ORF9b in complex with the cytosolic domain of human TOM70 has already been published in December 2020 (Gordon et al., 2020, Science). This paper is cited in the introduction as a reference for the interaction between the two proteins (p.4, line 68), but the EM-structure is only mentioned for the first time in the discussion (p. 11, lines 232ff). I understand that the authors may have perceived full acknowledgement of the previously published cryo-EM structure as potentially diminishing the impact and novelty of their crystal structure. Nevertheless, the EM-structure forms a considerable part of the context that needs to be introduced to the reader before the results are presented. Essentially, the crystal structure confirms the findings of Gordon et al., which is to be expected since the authors used the EM model to solve the crystal structure via MR. Therefore, a thorough comparison of these two structures should be included into the results section rather than into the discussion.
2. The introduction is very short, neither an objective to solve the complex structure nor a brief summary of the results are given.
3. In the section stating the author contributions, only four of the six authors are mentioned (lines 282-285). Being crystallographers at the Paul Scherrer Institute at the SLS (Villigen, Switzerland), I would expect Vincent Olieric and Meitian Wang to have contributed to the

crystallographic part of the manuscript. However, as stated in the methods section (p. 16, lines 336ff.), data were collected at the Shanghai synchrotron radiation facility (SSRF). This needs to be clarified.

Minor point:

3. The wording "... pushes the bridging helix to the left and drags it further downward ..." (p. 8, lines 155-156) implies an intrinsic default orientation of the protein. It would be better to describe motions as relative changes rather than referring to distinct directions.

Reviewer #2 (Remarks to the Author):

This is a solid and timely manuscript describing the crystal structure of a Sars-CoV-2 protein, orf9b, interacting with a human factor, Tom70. The excitement associated with this particular interaction is that it has been shown that orf9b targeting mitochondria and the MAVS/TRAF3/TRAF6 signalosome to suppress innate immune signaling, and the interaction with Tom70 may provide a mechanism for this phenomenon.

Although the structural findings of the paper are not novel in light of a recent manuscript in Science (Gordon et al 2020) describing the same structure determined by EM, this structure is at significantly higher resolution adding new details about the interaction that may be important if a drug targeting strategy is taken. The structural work is solid and the peptide is placed unambiguously. The statistics for the structure are provided and do not raise any red flags.

The more novel aspects of the manuscript are related to the findings of the affinity of binding of the Orf9b peptide and especially the interaction between the binding of orf9b and Hsp90. These findings confirm and expand on models that were put forth without evidence in the Gordon et al. manuscript. In particular, the ITC is rigorously done and adds a great deal of understanding to the current model of orf9b function, as well as confirming the underlying hydrophobic nature of the interaction.

One major question is: does binding of EEVD motif affect binding of the orf9b "C-peptide", positively or negatively. Also is binding of the EEVD motif blocked by the "C-peptide"? If not, the C-peptide might not be a very good model of orf9b binding.

minor points. The manuscript has some awkward phrasing that could benefit from careful editing. Some of these are listed below, but there are likely additional corrections to make.

42 delete "the"

52 delete "" on misleading

60 "presents" should be "is present"

71 more introduction to the role of TOM70 and Hsp90 in immune signaling would be appropriate here. If space is limited, less words can be devoted to the introduction of SARS. Everyone knows about SARS at this point!

79 this sentence is repetitive

106 "was" should be "were"

107 "went missing" - maybe meant "was disordered?"

133 Sentence starting "Despite..." is awkward, please rephrase

155 "the entering of orf9b" is awkward

Figure 2 the omit map in the insert is hard to see, if contrast was increased it would be improved.

Figure 3B, why is the bottom of the complex cut off?

Point-to-point responses to reviewers' comments.

REVIEWERS' COMMENTS

Reviewer #1 (Remarks to the Author):

The identification and characterization of interactions between proteins of the pandemic SARS-CoV-2 coronavirus and proteins of its human host constitute important groundwork for the development of potential drugs to treat COVID-19, the disease caused by SARS-CoV-2.

In the presented manuscript, Gao et al. describe the crystal structure of a binary complex formed by the cytosolic portion of the human mitochondrial translocase TOM70 and the protein encoded by open reading frame 9b of SARS-CoV-2 (ORF9b) to a resolution of 2.2 Å. They compare their complex structure to a previously solved structure of substrate-free TOM70 from yeast and find that binding of ORF9b may stabilize TOM70 in a rigid conformation. Additionally, the authors perform ITC experiments using TOM70 or TOM70/ORF9b with synthetic peptides derived from ORF9b and from the chaperone Hsp90 as binding partners. The peptides bind to spatially distinct binding sites located in the C-terminal domain (CTD) and in the N-terminal domain (NTD), respectively. They find that the binding constant of the Hsp90-peptide and the preformed TOM70/ORF9b complex is strongly reduced compared to that of Hsp90-peptide and TOM70 alone. From this, the authors conclude that binding of ORF9b to TOM70 suppresses binding of Hsp90-peptide to the NTD of TOM70 allosterically. Furthermore, the authors determine the binding constants of several point mutants of TOM70 to ORF9b and essentially confirm the interaction interface observed in their crystal structure.

The study by Gao et al. reports a highly relevant crystal structure of the complex between the accessory protein ORF9b of SARS-CoV-2 and the host protein TOM70 that may have implications for future therapeutic approaches. The technical quality of the data seems to be generally high and the presented data support the conclusions of the manuscript. However, I have serious reservations to recommend the manuscript for publication in its current state.

Major points:

1. It is not mentioned that a cryo-EM structure of ORF9b in complex with the cytosolic domain of human TOM70 has already been published in December 2020 (Gordon et al., 2020, Science). This paper is cited in the introduction as a reference for the interaction between the two proteins (p.4, line 68), but the EM-structure is only mentioned for the first time in the discussion (p. 11, lines 232ff). I understand that the authors may have perceived full acknowledgement of the previously published cryo-EM structure as potentially

diminishing the impact and novelty of their crystal structure. Nevertheless, the EM-structure forms a considerable part of the context that needs to be introduced to the reader before the results are presented. Essentially, the crystal structure confirms the findings of Gordon et al., which is to be expected since the authors used the EM model to solve the crystal structure via MR. Therefore, a thorough comparison of these two structures should be included into the results section rather than into the discussion.

Response:

We agree with the reviewer on this point. We added a paragraph (page 5, line 88-105) to introduce the EM-structure reported by Gordon et al in the Introduction section. We moved the comparison of the EM and X-ray structures to the Result section (Page 15, line 310-329 and page 16, line 330-333).

2. The introduction is very short, neither an objective to solve the complex structure nor a brief summary of the results are given.

Response:

We expanded the introduction, objective and brief summary of the result are added, please check page 4-7.

3. In the section stating the author contributions, only four of the six authors are mentioned (lines 282-285). Being crystallographers at the Paul Scherrer Institute at the SLS (Villigen, Switzerland), I would expect Vincent Olieric and Meitian Wang to have contributed to the crystallographic part of the manuscript. However, as stated in the methods section (p. 16, lines 336ff.), data were collected at the Shanghai synchrotron radiation facility (SSRF). This needs to be clarified.

Response:

We thank the Referee for pointing this omission. Vincent Olieric and Meitian Wang contributed to the crystallographic part with attempts at solving the TOM70/orf90 structure by the native-SAD method. They helped with the structure analysis and the revision of the manuscript and offered many valuable suggestions. We added their contribution in the revised manuscript, page 26.

Minor point:

3. The wording "... pushes the bridging helix to the left and drags it further downward ..." (p. 8, lines 155-156) implies an intrinsic default orientation of the protein. It would be better to describe motions as relative changes rather than referring to distinct directions.

Response:

We rewrote the sentence by describing relative motion changes. Please check the sentence in the top paragraph of page 11 (Line 225-227).

Reviewer #2 (Remarks to the Author):

This is a solid and timely manuscript describing the crystal structure of a Sars-CoV-2 protein, orf9b, interacting with a human factor, Tom70. The excitement associated with this particular interaction is that it has been shown that orf9b targeting mitochondria and the MAVS/TRAF3/TRAF6 signalosome to suppress innate immune signaling, and the interaction with Tom70 may provide a mechanism for this phenomenon.

Although the structural findings of the paper are not novel in light of a recent manuscript in Science (Gordon et al 2020) describing the same structure determined by EM, this structure is at significantly higher resolution adding new details about the interaction that may be important if a drug targeting strategy is taken. The structural work is solid and the peptide is placed unambiguously. The statistics for the structure are provided and do not raise any red flags.

The more novel aspects of the manuscript are related to the findings of the affinity of binding of the Orf9b peptide and especially the interaction between the binding of orf9b and Hsp90. These findings confirm and expand on models that were put forth without evidence in the Gordon et al. manuscript. In particular, the ITC is rigorously done and adds a great deal of understanding to the current model of orf9b function, as well as confirming the underlying hydrophobic nature of the interaction.

One major question is: does binding of EEVD motif affect binding of the orf9b "C-peptide", positively or negatively. Also is binding of the EEVD motif blocked by the "C-peptide"? If not, the C-peptide might not be a very good model of orf9b binding.

Response:

To address this question, we performed two more experiments to investigate whether the N-peptide (EEVD motif) affect the binding of the C-peptide to TOM70? Whether the C-peptide act like the intact orf9b protein in binding with TOM70?

- (1) We titrated the N-peptide to the hTOM70/C-peptide complex. The C-peptide was pre-incubated with the unliganded hTOM70 (molar ratio 3:1) to allow complex formation; the N-peptide was then added to the complex. We observed ~13 folds reduction in binding affinity of the N-peptide to hTOM70/C-peptide complex (Fig. S3B) comparing to that between the N-peptide and the unliganded hTOM70. It is clear that the C-peptide was able to block the N-peptide binding like the intact orf9b, although to a lesser extent. The difference is likely attributed to significantly lower molecular mass of the C-peptide.
- (2) Conversely, we titrated the C-peptide to hTOM70/N-peptide complex. The N-peptide was pre-incubated with the unliganded hTOM70 (molar ratio 3:1) to allow complex formation, and the C-peptide was then added. The binding of the C-peptide to hTOM70/N-peptide complex was also an endothermic process (like the interaction between C-peptide to the unliganded hTOM70) with the KD of 0.42 μ M (Fig. S3C). This

result indicates the N-peptide has little effect on binding affinity of the C-peptide to hTOM70.

We added these results in(page 13,line 276-284 and page 14, line 285-292) minor points. The manuscript has some awkward phrasing that could benefit from careful editing. Some of these are listed below, but there are likely additional corrections to make.

Response:

We used a professional English editing service to improve the writing. The editing certificate is available to view online at the following URL.

<https://www.bioedit.com/digital-certificate/view/31bf6921e1d8a3686a9d056d811330ac0903f251>

42 delete "the"

Response:

deleted.

52 delete "" on misleading

Response:

deleted.

60 "presents" should be "is present"

Response:

Fixed.

71 more introduction to the role of TOM70 and Hsp90 in immune signaling would be appropriate here. If space is limited, less words can be devoted to the introduction of SARS. Everyone knows about SARS at this point!

Response:

We added introduction of the role of TOM70 and Hsp90 in immune signaling at the bottom of page 4(line75-84)and page 5(line85-87)

79 this sentence is repetitive

Response:

Sentence deleted.

106 "was" should be "were"

Response:

Changed.

107 “went missing” - maybe meant “was disordered?”

Response:

We replace the words with disordered.

133 Sentence starting “Despite...” is awkward, please rephrase

Response:

We rewrote the sentence to: “While the majority of those polar residues on the surface of the CTD-pocket are conserved, the counterpart of E549 in yeast TOM70/TOM71 is isoleucine”

155 “the entering of orf9b” is awkward

Response:

We changed it to “the binding of orf9b”

Figure 2 the omit map in the insert is hard to see, if contrast was increased it would be improved.

Response:

We provide a new Figure 2, in which the contrast of the omit map has been increased to improve visibility.

Figure 3B, why is the bottom of the complex cut off?

Response:

We provide new Fig.3 to fix this problem.